# Genetic Analysis of the Grapevine GATA Gene Family and Their Expression Profiles in Response to Hormone and Downy Mildew Infection

**Tingting Chen** [1,2,3,†], **Jing Peng** [1,3,†], **Meijie Li** [1,3], **Mengru Dou** [1,3], **Yan Lei** [4], **Yuejing Wang** [1,3] and **Yan Xu** [1,3,*]

1  State Key Laboratory of Crop Stress Biology in Arid Region, College of Horticulture, Northwest A&F University, Yangling 712100, China; ctt1991@nwafu.edu.cn (T.C.); ppjing@nwafu.edu.cn (J.P.); meijie.li@supagro.fr (M.L.); dmr@nwafu.edu.cn (M.D.); wangyj@nwsuaf.edu.cn (Y.W.)
2  College of Agricultural Science, Xi Chang University, Xichang 615000, China
3  Key Laboratory of Horticultural Plant Biology and Germplasm Innovation in Northwest China, Ministry of Agriculture, Northwest A&F University, Yangling 712100, China
4  Fruit Research Institute, Fujian Academy of Agricultural Sciences, Fuzhou 350013, China; lxmy2010@163.com
*  Correspondence: yan.xu@nwsuaf.edu.cn
†  These authors contributed equally to this work.

**Abstract:** Grapevine (*Vitis. vinifera* L.) is one of the most economically important fruit crops throughout the world. However, grape production is increasingly impacted by numerous diseases, including downy mildew, caused by the oomycete *Plasmopara viticola*. In grapevine and other plants, members of the GATA family of transcription factors play key roles in light and phytohormone signaling. However, little is known about their potential roles in biotic defense responses. As a first step, we identified 27 GATA transcription factors in grapevine and defined their transcriptional responses to three biotic stress-related phytohormones (SA, MeJA, and BR) in callus cells, and challenge with *P. viticola* in a downy mildew-sensitive cultivar, *V. vinifera* 'Pinot noir', and a resistant cultivar, *V. piasezkii* 'Liuba-8'. Many of the VvGATA genes had higher expression at 0.5 h after hormones treatments. Moreover, a group of VvGATAs was dramatically induced in 'Liuba-8' at 24 post infection by *P. viticola*. However, the same genes were significantly repressed and showed low expression levels in 'Pinot noir'. Additionally, VvGATA27 was located in the nucleus and had transcriptional activity. Taken together, the study identified the GATA full gene families in grapes on phylogenetic analysis and protein structure. Moreover, this study provided a basis for discussing the roles of VvGATAs in response to hormones and *P. viticola* infection. Our results provided evidence for the selection of candidate genes against downy mildew and lay the foundation for further investigation of VvGATA transcription factors.

**Keywords:** grapevine; GATA transcription factors; *Plasmopara viticola*; innate immunity

## 1. Introduction

*Vitis* species and cultivars vary in resistance to *Plasmopara viticola*. Generally, European grape (*Vitis. vinifera* L.) cultivars widely are highly susceptible to *P. viticola*, while American and Asian *Vitis* species have varying degrees of resistance [1–4]. For example, *V. vinifera* cultivars such as 'Pinot noir' are susceptible to *P. viticola*, while in North America, *V. rupestris* and *V. riparia* are highly resistant; even *Muscadinia rotundifolia* is immune [1,4,5]. In our previous studies, Chinese wild grapes have been reported for its high resistance to powdery mildew [6]; especially *V. piasezkii* showed strong resistance [7]. Subsequently, our lab studied the response to *P. viticola* in three Chinese wild grapes (*Vitispseudoreticulata* Baihe-35-1, *Vitis davidii* var. Cyanocarpa Langao-5, and *Vitis piasezkii* Liuba-8) and European cultivated variety (*V. vinifera* cv. Pinot noir), showed Chinese wild grapevine 'Liuba-8' (LB, *V. piasezkii*) is highly resistant, while *vinifera* cv. Pinot noir is susceptible at the histological

level [1,4]. However, genetic resistance often fails to overcome *P. viticola* due to limited knowledge of the molecular biology and interaction mechanism of plants and pathogens.

Plant growth is constantly challenged by biotic and abiotic pressures [8]. Plants respond to such challenges by deploying a cadre of transcriptional regulators that activate key genes and pathways important for defense [9–11]. GATA transcription factors comprise an evolutionarily conserved family of zinc finger that regulate genes through binding to a defined DNA cis-element ((T/A) GATA (A/G)) in gene regulatory regions [12,13]. In plants, GATA factors have been shown to be involved in various aspects of growth and development, as well as response to drought, heat, and cold stress [14]. For example, the GATA3-family transcription factors play a critical role in cell proliferation and differentiation, and abnormal expression of the GATA3-type gene (HAN) results in loss of meristematic activity and normal growth patterns in *Arabidopsis* [15]. In *Arabidopsis*, the GATA transcription factors GNC and CGA1 play an important role in the growth, development, and division of chloroplasts and act to postpone flowering in response to low temperatures [16]. The GATA transcription factors PdGNC from poplar, as well as OsGATA1 and OsGATA12 from rice, regulate chloroplast structure, photosynthesis, and plant growth and development [17–20]. In bamboo, 31 GATA genes were identified, which provided insight into engineering plant height [21]. Bhardwaj et al. [22] identified 29 GATA transcription factors in the oilseed crop *Brassica juncea* and found that five of them were downregulated under drought stress, and two of them were upregulated under high-temperature stress. Interestingly, GATA2 from *Arabidopsis* directly regulates genes involved in both light- and brassinosteroid-mediated signaling pathways [23].

Although plant GATA transcription factors have been extensively studied in the context of growth and development, little is known about their potential roles in plant immunity. Zhang et al. [24] identified a GATA gene from *Cucumis hystrix* as a candidate resistance gene for cucumber downy mildew and powdery mildew. Subsequently, the wheat TaGATA1 gene was found to be upregulated during the host resistance response to *Rhizoctonia cerealis* [25]. The grape VdGATA2 could activate the oxygen species pathway to enhance powdery mildew resistance [26].

A previous analysis provided insight into the evolutionary relationships and genomic organization for 19 GATA family members in *V. vinifera* and demonstrated that a subset of VvGATA genes is transcriptionally regulated in response to light and phytohormones [27]. In this study, we expanded this characterization to include all of the 27 annotated VvGATA genes in the grapevine and also evaluated their potential as transcription factors in the response of the grapevine to *P. viticola*. Furthermore, we characterize a subfamily IV GATA transcription factor, VvGATA27, which may be a key gene in response to pathogen infection. These analyses offer new insights into the roles of VvGATA factors in response to hormones and downy mildew infection in grapes and lay the foundation for further studies of VvGATA transcription factors.

## 2. Materials and Methods

### 2.1. Identification of GATA Family Genes in Grapevine

GATA transcription factors in grapevine were identified (1) as cataloged in the Plant Transcription Factor Database (V5.0; Jin et al. [28]; http://planttfdb.gao-lab.org/; accessed on 26 December 2019); (2) by HMMER [29] using the GATA zinc finger (PF00320) as indexed in Pfam (http://pfam.xfam.org/; accessed on 26 December 2019) as a query pattern, and used to blast the genome-wide of GATA motif in *V. vinifera* genome database (12×) (https://www.genoscope.cns.fr/externe/GenomeBrowser/Vitis/; accessed on 26 December 2019) with default E-values (<1.0); and (3) by BLAST, using all annotated Arabidopsis GATA genes as queries with the NCBI nr database (https://www.ncbi.nlm.nih.gov/; accessed on 26 December 2019). Identified protein sequences were filtered to remove sequences lacking a GATA-type zinc-finger motif as identified using the online software SMART (http://smart.embl-heidelberg.de/; accessed on 26 December 2019) and PfamScan (https://www.ebi.ac.uk/Tools/pfa/pfamscan/; accessed on 26 December 2019). Theoretical

isoelectric points and molecular masses were calculated using an online tool (https://web.expasy.org/compute_pi/; accessed on 26 December 2019).

### 2.2. Phylogenetic Analysis, Protein Structural Analysis, and Identification of Promoter Cis-Elements

The phylogenetic tree was constructed using the neighbor-joining, and minimal evolution methods in MEGA 5.0 with 1000 bootstrap replicates. Conserved protein motifs were predicted by MEME (http://meme-suite.org/tools/meme; accessed on 26 December 2019) using default parameters (putative motifs number: 20; accessed on 26 December 2019), with a maximum E-value cutoff of $1 \times 10^{-30}$. Predicted motifs were annotated using the InterProScan database (http://www.ebi.ac.uk/interpro/search/sequence/; accessed on 26 December 2019). Exon/intron organization of VvGATA family genes was illustrated using the Gene Structure Display Server (http://gsds.gao-lab.org/index.php; accessed on 26 December 2019). Cis-elements in the promoter regions of the VvGATA genes were identified with PLANT CARE (http://bioinformatics.psb.ugent.be/webtools/plantcare/html/; accessed on 26 December 2019).

### 2.3. Plant Material, Fungal Materials, and Inoculation

The inoculated leaves, *V. vinifera* 'Pinot noir' (PN) and V. piasezkii 'Liuba-8′ (LB), were accessed in the Vitis germplasm collection of Northwest A & F University, Yangling, Shaanxi, China. Similarly, the leaves with a downy mildew layer were collected from *Vitis* germplasm collection. Firstly, infected leaves were washed three times with sterilized ddH$_2$O and then transferred into trays covered with sterile moist filter papers. In addition, humidity and a dark environment benefit *P. viticola* speculation formation. When the back of leaves appeared with fresh downy mildew, we used a sterilized soft brush to collect fresh sporulation in sterile distilled water and filter with three layers of sterilization gauzes. The concentration of the sporangia suspension was adjusted to $5 \times 10^4$ sporangia/mL under optical microscope. Then, the adjusted sporangia suspension were sprayed on the abaxial leaf surface, and the control group was sprayed with sterile ddH$_2$O. The inoculated leaves were cultivated in an incubator (temperature: $23 \pm 1$ °C; humidity: 90%, and photoperiod: 16 h light/8 h dark). Samples were taken at 5 time points: 0, 12, 24, 48, 96, and 120 hpi, with 0 hpi as the control samples. The collected samples were immediately frozen in liquid nitrogen and stored at $-80$ °C for later use. All treatments were carried out with three biological replicates from three independent leaves.

### 2.4. Callus Cells Treatments

Callus cells from cv. Thompson seedless were prepared as described in Wang et al. [30]. For hormone treatment, callus cells were transferred to fresh growth medium and harvested about one week later when in the logarithmic growth phase. Cells were resuspended in 50 mL of hormone-free B5 medium (B5 basal medium supplemented with 1.0 mg/L 2,4-D, 0.5 mg/L 6-BA and 0.2% PVP). Hormone was added (100 μM salicylic acid (SA), 100 μM methyl jasmonate (MeJA), or 100 μM brassinosteroid (BR)) according to the methods described in Sun et al. [31].

### 2.5. RNA Extraction and Expression Pattern Analysis

For analysis of VvGATA gene expression in response to hormone treatment or P. viticola infection, RNA was purified from frozen samples using a commercial kit (Plant RNA; OMEGA Bio-tek, Norcross, GA, USA). The quality of total RNA was assessed by electrophoresis on 1% agarose gels, and RNA concentration was determined using a spectrophotometer (NanoPhotometer®; IMPLEN, Westlake Village, CA, USA). Total RNA was treated with DNAse (gDNA Wiper Mix; Vazyme, Nanjing, China) prior to cDNA synthesis. Total RNA (500 ng) was subjected to reversed transcription using the HiScript® IIReverse Transcriptase Kit (Vazyme, Nanjing, China) according to the manufacturer's instructions. Oligonucleotide primers for RT-PCR assays were designed for specificity to

each gene based on sequence information (Table S1. Quantitative PCR was carried out as described in a previous study [32]. The VvACTIN 1 (AY680701) gene was used as an internal reference as described by Li et al. [32]. The relative expression levels of VvGATA genes were calculated using the $2^{-\Delta\Delta c(t)}$ method. Each reaction was performed in three biological and technical replicates for each sample. All data analyses were conducted using SPSS 20 Software and used one-way ANOVA analysis to conduct the significant difference ($p < 0.05$). The relative expression values were log2 transformed. Average linkage clustering was used to cluster genes in cluster limits 3.0 and visualized using MEV software.

### 2.6. Confocal Imaging and Transcriptional Assay

For visualization of the expression of the VvGATA27-GFP fusion protein or GFP and NLS-mCherry as a nuclear marker, protoplasts were prepared from callus cells from cv. Thompson seedless and subjected to polyethylene glycol (PEG)-mediated transformation as described by Martin et al. [33]. The amino acid sequence of the SV40 T large antigen NLS is PKKKRKV. Protoplasts were photographed after incubation for 20–24 h in a weak light at 25 °C. Fluorescence of GFP and m-Cherry was excited at 488 nm and 561 nm, respectively, by confocal microscopy (LEICA TCS SP8, Wetzlar, Germany). Transcriptional activation assays were carried out using the GAL4-based YeastmakerTM Yeast Transformation System 2 as described in the user manual (Clontech, San Jose, CA, USA). The VvGATA27 factor open reading frame was amplified by PCR from *V. vinifera* cDNA and cloned into the prey vector pGBK-T7. The pGBKT7-VvGATA27 constructions and pGBK-T7 (as negative control) were introduced into the yeast strain Y2H and grown on selective media (SD/-Trp) at 28 °C for 3 days. Colonies were then assessed for growth on selective single dropout (SD/-Trp) media in the presence of Aba and X-α-Gal for 3 days.

## 3. Results

### 3.1. Identification of GATA Transcription Factor Genes in V. vinifera

To comprehensively identify putative GATA transcription factor in grapevine, we initially retrieved all grapevine sequences cataloged as GATA transcription factors. A homology-based search was then carried out using the *V. vinifera* genome database, with the GATA zinc finger indexed in Pfam as a query pattern, and the NCBI nr database and BLAST, using all 29 annotated *Arabidopsis* GATA genes as queries. These combined approaches resulted in the identification of 27 distinct sequences, which were designated GATA1-GATA27. Of these sequences, only two (VvGATA11 and VvGATA12, default blast E-value 0.02 and 0.49, respectively) lacked clear orthology with *Arabidopsis* GATA transcription factors. A total of 19 of these 27 genes were previously identified as GATA factors [27]. The length of the encoded proteins ranged from 104 to 548 aa, and the isoelectric points and molecular masses ranged from 4.78 (VvGATA26) to 10.2 (VvGATA14) and 12 kDa (VvGATA12) to 60 kDa (VvGATA27), respectively. Table 1 lists the gene locus IDs, GenBank accession numbers, genome location, amino acid length, subfamily group, conserved protein motifs, and HMMER/BLAST E-values. Additional information, including ORF length, isoelectric point, and molecular mass, and orthology with *Arabidopsis* genes is listed in Supplementary File Table S2.

**Table 1.** Detailed information of identified GATA factors.

| Gene Name | Gene Cribi ID | Genome Location | GeneBank Accession No. | Motif | | | | Length (aa) | E-Value | Used Name (NCBI, Vitis Vinifera) |
| --- | --- | --- | --- | --- | --- | --- | --- | --- | --- | --- |
| | | | | Subgroup | znF-GATA | Tify | Another | | | |
| VvGATA1 | VIT_03s0038g00490 | chr3, 452898..453933 | XM_002274836.3 | I | 1 | | | 317 | $4.33 \times 10^{-13}$ | GATA transcription factor 5 |
| VvGATA2 | VIT_03s0038g00580 | chr3_random, 512571..559934 | XM_019217517.1 | III | 1 | 1 | CCT | 353 | $4.10 \times 10^{-15}$ | GATA transcription factor 19 |
| VvGATA3 | VIT_03s0038g00480 | chr3_random, 452201..461888 | XM_002270325.4 | III | 1 | 1 | CCT | 302 | $6.56 \times 10^{-13}$ | GATA transcription factor 24 |
| VvGATA4 | VIT_04s0008g01290 | chr4, 1062865..1063577 | XM_002279247.3 | II | 1 | | | 306 | $4.49 \times 10^{-20}$ | GATA transcription factor 22 |
| VvGATA5 | VIT_04s0023g01840 | chr4, 18421693..18422139 | XM_010650837.2 | II | 1 | | | 249 | $2.00 \times 10^{-17}$ | GATA transcription factor 18 |
| VvGATA6 | VIT_04s0023g02880 | chr4, 19468180..19469275 | XM_002272726.3 | I | 1 | | | 338 | $1.28 \times 10^{-15}$ | GATA transcription factor 5 |
| VvGATA7 | VIT_04s0008g03270 | chr4, 2730051..2730368 | XM_002283709.3 | I | 1 | | | 342 | $1.60 \times 10^{-16}$ | GATA transcription factor 9-like |
| VvGATA8 | VIT_05s0051g00450 | chr5, 11067051..11068106 | XM_010651913.1 | I | 1 | | | 258 | $5.13 \times 10^{-15}$ | GATA transcription factor 1 |
| VvGATA9 | VIT_05s0077g01450 | chr5, 1164009..1164539 | XM_010651243.2 | II | 1 | | | 153 | $6.12 \times 10^{-13}$ | GATA transcription factor 16 |
| VvGATA10 | VIT_206s0004g01275 | chr6, 1521644..1522690 | XM_019220522.1 | II | 1 | | | 162 | $5.08 \times 10^{-19}$ | GATA transcription factor 12-like |
| VvGATA11 | none | chr6, 1526283..1528520 | XM_010653013.2 | II | 1 | | | 171 | $6.22 \times 10^{-14}$ | GATA transcription factor 13 |
| VvGATA12 | VIT_06s0004g01280 | chr6, 1535752..1535895 | XM_019220525.1 | II | 1 | | | 104 | $1.50 \times 10^{-16}$ | GATA transcription factor 23-like |
| VvGATA13 | VIT_06s0004g02740 | chr6, 3428913..3432090 | XM_010652872.2 | I | 2 | | | 464 | $3.52 \times 10^{-13}$ | GATA transcription factor 8 |
| VvGATA14 | VIT_207s0005g01085 | chr7, 3630384..3630512 | XM_010653922.2 | II | 1 | | | 140 | $2.71 \times 10^{-14}$ | GATA transcription factor 15 |
| VvGATA15 | VIT_08s0007g07550 | chr8, 21037563..21038886 | XM_002282189.3 | I | 1 | | | 299 | $3.63 \times 10^{-15}$ | GATA transcription factor 2 |
| VvGATA16 | VIT_09s0054g00440 | chr9, 20974328..20988907 | XM_002263671.4 | III | 1 | 1 | CCT | 299 | $2.32 \times 10^{-13}$ | GATA transcription factor 24 |
| VvGATA17 | VIT_09s0002g03750 | chr9, 3439127..3440408 | XM_002283992.3 | I | 1 | | | 329 | $3.95 \times 10^{-16}$ | GATA transcription factor 9 |
| VvGATA18 | VIT_11s0016g02210 | chr11, 1818033..1819175 | XM_002282137.3 | II | 1 | | | 309 | $2.72 \times 10^{-19}$ | GATA transcription factor 21 |
| VvGATA19 | VIT_12s0121g00120 | chr12, 11859275..11882585 | XM_019223388.1 | IV | 1 | | | 216 | $4.00 \times 10^{-15}$ | GATA transcription factor 27 |
| VvGATA20 | VIT_14s0066g00150 | chr12, 11878739..11880211 | FN596758.1 | IV | 1 | 1 | BET | 259 | $4.00 \times 10^{-12}$ | unnamed protein product |
| VvGATA21 | VIT_13s0019g04390 | chr13, 5760956..5762854 | XM_002273466.4 | I | 1 | | | 340 | $6.51 \times 10^{-17}$ | GATA transcription factor 8 |
| VvGATA22 | VIT_14s0060g01520 | chr14, 1203496..1204229 | XM_002278333.4 | II | 1 | | | 125 | $8.46 \times 10^{-16}$ | GATA transcription factor 16 |
| VvGATA23 | VIT_14s0066g00150 | chr14, 26717092..26720948 | XM_010662532.2 | II | 1 | 1 | | 367 | $5.30 \times 10^{-15}$ | GATA transcription factor 7 |
| VvGATA24 | VIT_15s0021g02510 | chr15, 13505714..13506649 | XM_002277923.3 | I | 1 | | | 270 | $1.45 \times 10^{-14}$ | GATA transcription factor 4 |
| VvGATA25 | VIT_18s0001g07720 | chr18, 6040264..6057788 | XM_002283717.3 | III | 1 | | CCT | 294 | $2.52 \times 10^{-14}$ | GATA transcription factor 25 |
| VvGATA26 | VIT_18s0001g07730 | chr18, 6060621..6094190 | XM_010666122.2 | III | 1 | | CCT | 368 | $8.90 \times 10^{-17}$ | GATA transcription factor 24 |
| VvGATA27 | VIT_200s2393g00010 | chrUn, 41800730..41802661 | AM449204.2 | IV | 1 | | ASXH | 542 | $5.60 \times 10^{-16}$ | GATA transcription factor 26 |

### 3.2. Phylogenetic Analysis, Conserved Motifs, and Structural Features of the V. vinifera GATA Gene Family

We next conducted a phylogenetic analysis to clarify the relationships among all putative GATA transcription factors from grapevine and Arabidopsis (Figure 1; Supplementary File Table S3). According to the four subgroups of GATA factors assigned from Arabidopsis, we identified 9 grapevines GATA factors in subgroup I, 10 genes in subgroup II, 5 genes in subgroup III, and 3 genes in subgroup IV. The eight grapevine GATA factors that had not previously been identified by Zhang et al. [27] were distributed among subgroups I, II, and IV (Figure 1). Two of the previously unidentified factors, VvGATA20 and VvGATA27, were included in a well-defined (99% of bootstrap replicates) clade with the previously identified VvGATA19 and two *Arabidopsis* genes, At4G17570 and At5G47140. An amino acid sequence alignment revealed that VvGATA27 was particularly closely related to At4G17570c (Figure 1 and Figure S2). In addition, VvGATA27 contained specific conservative motif ASXH belonging to subgroup IV as reported in *Arabidopsis* and pepper [14,34]. To characterize the VvGATA proteins further, conserved motifs were identified [35]. A total of seven conserved motifs were identified among the 27 GATA protein sequences (Figure 2; Supplementary Materials Figure S3). factors within the same subgroup were generally found to exhibit the same number of exons and similar intron-exon organization (Figure S3).

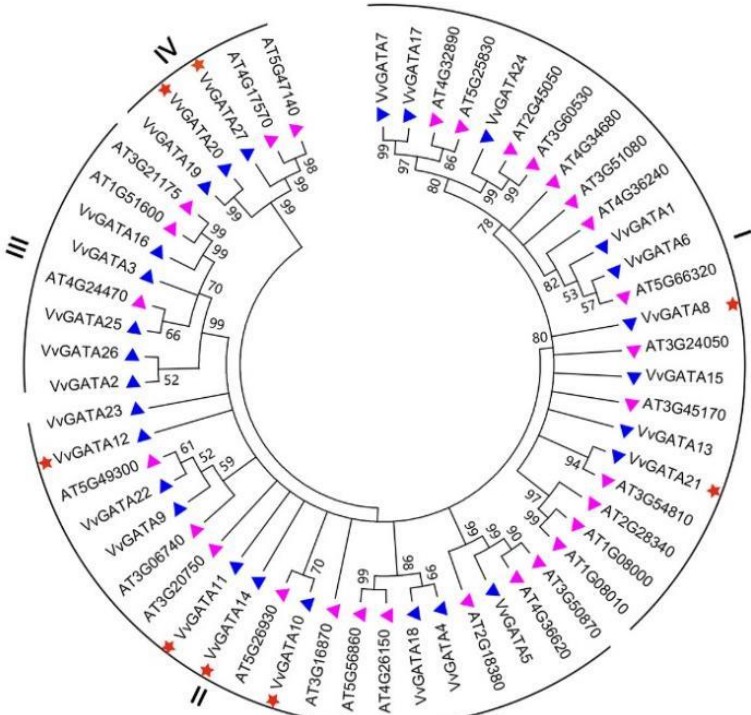

**Figure 1.** Phylogenetic analysis of GATA proteins from grapevine and *Arabidopsis*. GATA genes from grapevine and *Arabidopsis* are marked in blue and pink, respectively. The eight newly identified GATA genes are marked by a red star. The four subfamilies are indicated in Roman numerals. The phylogenetic tree was constructed by MEGA 5.0 using the neighbor-joining method with 1000 bootstrap replicates.

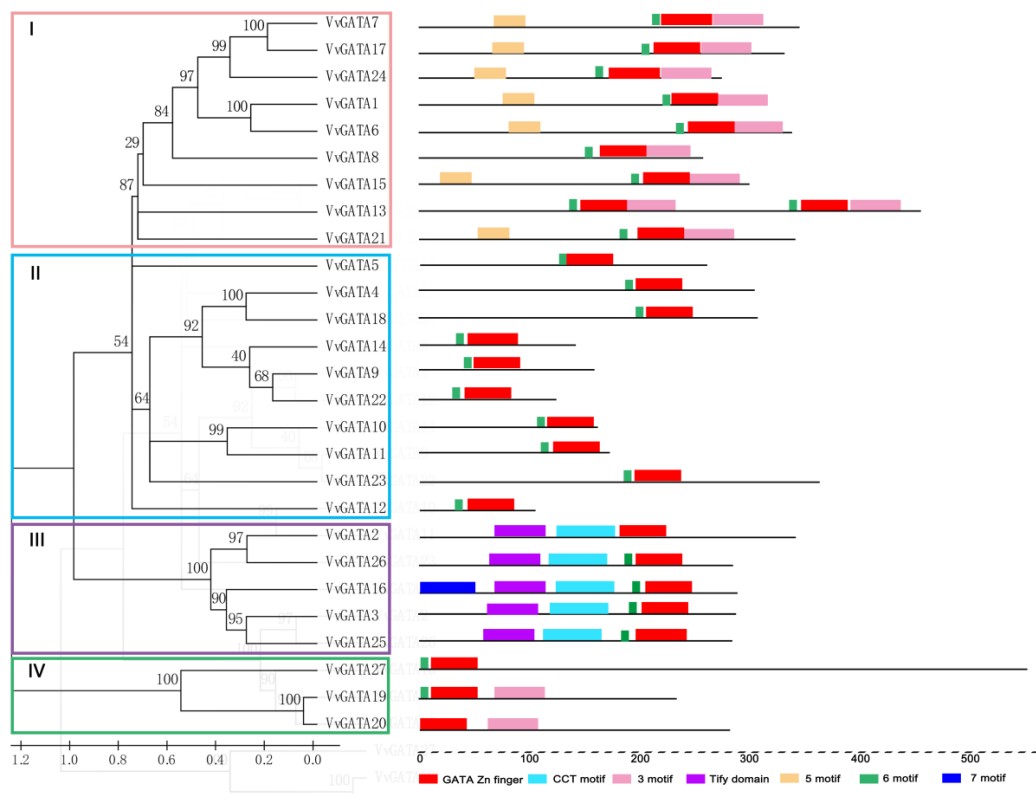

**Figure 2.** Structural characteristics of the GATA family transcription factors in grape. Phylogenetic analysis of VvGATA proteins. On the left, Subfamilies (**I–IV**) are enclosed in red, blue, purple, and green rectangular boxes, respectively. On the right, Identification of conserved structural domains within VvGATA proteins. Conserved regions and domain identities were determined by MEME.

### 3.3. Cis-Elements in the Promoter Region of VvGATA Factors

To investigate the potential regulation of VvGATA factors by hormone or biotic stress pathways, the upstream 1500 bp promoter sequence for each VvGATA factor was analyzed for the presence of putative cis-acting elements. We identified 72 cis-elements, and numerous potential cis-elements were defined to respond to light, and the cis-element was distributed on each gene promoter (Table S4). Twenty cis-elements major involved in hormone and stress responses were identified on the promoter of 27 VvGATA factors (Figure 3, Table 2). Except for the light cis-elements, the cis-elements of MYC and MYB account for the biggest proportion in the promoter of VvGATA factors (Table 2). There were 17 cis-elements that were a response to hormones was observed across all VvGATA factors, except VvGATA2 and VvGATA1, respectively (Figure 3). In addition, various cis-elements related to abiotic stress (e.g., STRE, Myb biARE motif for gibberellin response, as-1/TCA-element for SA, and CGTCA-motif/JERE for JA, and TGA-element for auxin response) (Table 2). Regulatory elements involved in JA and ABA signaling (Mnding site/MBS for drought, LTR for cold) and defense responsiveness were identified in the promoter of VvGATA factors (Figure 3). Meanwhile, 17 VvGATAs contained cis-elements related to response to SA, and 27 VvGATAs have JA-responsive cis-elements (Table 2).

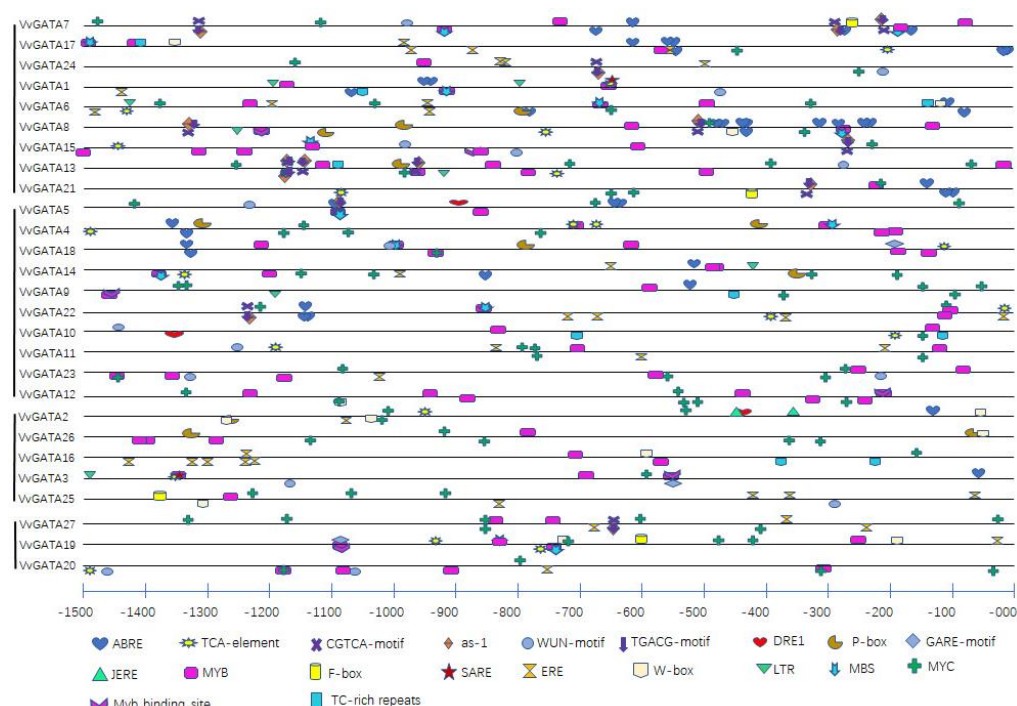

**Figure 3.** Promoter cis-element analysis of VvGATA factors. The upstream promoter sequence (1.5 kb) for all VvGATA factors was evaluated for number and position of cis-elements as predicted by Plant-CARE. The various elements are indicated by colored boxes and are shown in their corresponding positions on the promoter. Symbols presented above the line indicate the forward strand of DNA, while those below indicate the reverse strand.

**Table 2.** Common putative cis-elements in the promoter of the VvGATA genes.

| Cis-Element Name | Sequence | Response | Number | Gene Number |
|---|---|---|---|---|
| ABRE | ACGTG | Abscisic acid | 44 | 14 |
| TCA-element | CCATCTTTTT | Salicylic acid | 21 | 17 |
| CGTCA-motif | CGTCA | Methyl jasmonate | 15 | 9 |
| as-1 | TGACG | Salicylic acid | 15 | 9 |
| WUN-motif | AATTT(A)C | Mechanical injury | 16 | 13 |
| TGACG-motif | TGACG | Methyl jasmonate | 15 | 9 |
| DRE1 | ACCGAGA | Abscisic acid | 3 | 3 |
| P-box | CCTTTTG | Gibberellin | 11 | 8 |
| GARE-motif | TCTGTTG | Gibberellin | 3 | 3 |
| JERE | AGACCGCC | Methyl jasmonate | 2 | 1 |
| MYB | T(C)AAC | Methyl jasmonate | 89 | 27 |
| F-box | CTATTCTCATT | Abscisic acid, Light, Drought | 4 | 4 |
| SARE | TTCGACCATCTT | Salicylic acid | 2 | 2 |
| ERE | ATTTTAAA | Ethylene | 38 | 14 |
| W-box | TTGACC | Defense | 12 | 9 |
| LTR | CCGAAA | Low-temperature | 8 | 7 |
| MBS | CAACTG | Drought | 14 | 12 |
| MYC | CAA(T)G(T)TG(A) | Abscisic acid | 84 | 26 |
| Myb binding site | CAACAG | Drought | 7 | 7 |
| TC-rich repeats | TTC(T)TCT | Defense and stress | 9 | 7 |

Number, a total quantity of each cis-elements identified in the promoters of 27 GATA genes; Gene Number, a frequency of each cis-elements appeared in the promoters of 27 GATA genes.

### 3.4. Expression Profiles of the VvGATA Genes in Response to Hormones

To further evaluate the potential roles of the VvGATA factors in hormone signaling pathways, we monitored gene expression in cell suspensions subjected to treatment with

SA, MeJA, and BR (Figure 4). The expression data were placed in Additional File S1. For SA, the expression patterns could be divided into three groups (Figure 4A). Genes in group SA-1 showed a significant increase in expression at 3 h after treatment. Group SA-2 was upregulated at 12 h, but some were induced at 0.5 h followed by decreased expression thereafter, and others were downregulated at 0.5 h. In contrast, genes in group SA-3 exhibited a decline in expression from 0.5 to 3 h, then an increase in expression at 12 h after treatment. However, VvGATA1 and VvGATA3 were almost not induced under SA treatment. Under MeJA treatment, the majority of group JA-1 genes showed a slight increase at 0.5 h, except VvGATA9. Compared with 0.5 h under treatment, VvGATA17, VvGATA1, VvGATA9, and VvGATA18 showed low expression until 12 h. The remaining genes, in group JA-3, showed an increase at 0.5 h and decreased at 3 h after treatment, with varying responses thereafter. Meanwhile, VvGATA3, VvGATA4, and VvGATA7 were remarkably induced at 0.5 h. Three response groups could also be distinguished for BR treatment. In group BR-1, expression showed a bimodal response, with increases at 3 h and 12 h. Group BR-2, comprising 13 genes, most exhibited an early increase in expression, but VvGATA9, VvGATA2, and VvGATA5 were downregulated at 0.5 h after treatment. Group BR-3 genes increased from 0.5 to 3 h, then decreased immediately at 6 h after BR treatment (Figure 4C).

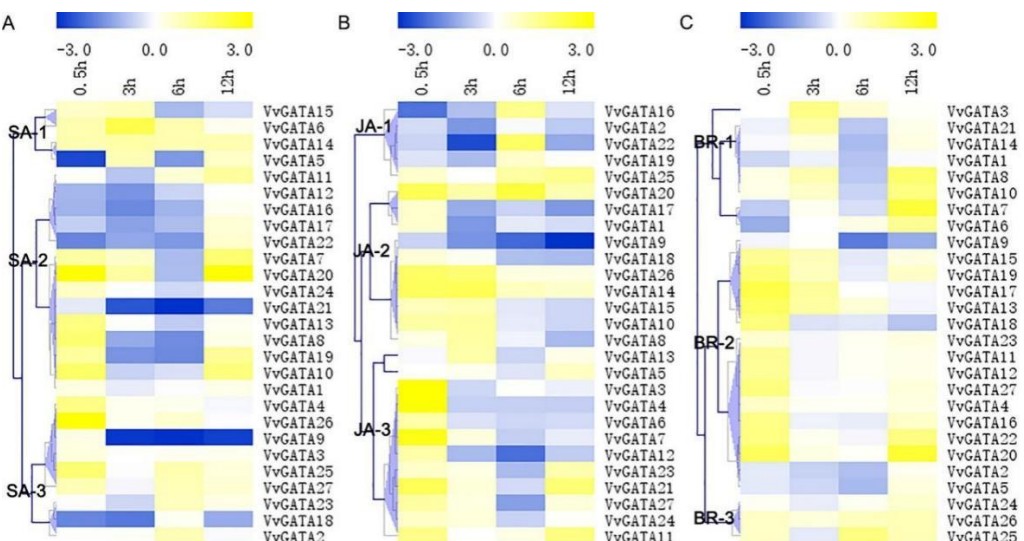

**Figure 4.** Heat map of transcript levels of the VvGATA family factors in response to different hormones. Expression of the 27 VvGATA factors was evaluated in callus cell suspensions derived from cv. Thompson seedless stems at different time points under SA (**A**), JA(MeJA) (**B**), or BR (**C**) treatments. The color scale represents expression relative to time 0, with dark blue representing strong down-regulation and yellow representing strong up-regulation. The grapevine actin gene VvACTIN (AY680701) was used as an internal reference. The relative expression levels were calculated using the $2^{-\Delta\Delta Ct}$ method, and the heat map was visualized using MeV software.

### 3.5. Expression Profiles of the VvGATA Genes in Response to P. viticola

To understand the potential roles of the VvGATA factors in resistance to powdery mildew, we assessed their expression following inoculation with *P. viticola*, both in the susceptible cultivar 'Pinot noir' (PN) and the resistant genotype *V. piasezkii* 'Liuba-8' (LB). Expression was evaluated as described by Liu et al. [1]. The expression data were placed in Additional File S1. In 'Pinot noir', most genes showed an increase in expression at 6 h post infection (hpi) and showed low expression levels at 12 and 24 h after inoculation. Additionally, VvGATA14 and VvGATA10 were strongly induced, and VvGATA1 was significantly downregulated by 6 hpi. This indicates that most VvGATA genes were quickly responsive to inoculation. Meanwhile, a subset of genes (VvGATA21, VvGATA20, VvGATA22, VvGATA6, VvGATA10, VvGATA4, VvGATA25, VvGATA24, VvGATA27, and

VvGATA14) was strongly induced at 48 hpi (Figure 5). Moreover, VvGATA11 was not sensitive to inoculation at any time point evaluated (Figure 5). The expression patterns in LB were different from those in PN in several respects. In LB, many of the VvGATA genes were initially non-responsive to inoculation at 6 hpi. Until 12 hpi to 24 hpi, a part of VvGATAs were repressed and showed extremely low expression levels. Especially, VvGATA9 was downregulated from 6 to 24 hpi. A group of VvGATA genes was upregulated by 12 to 24 hpi and dramatically induced at 24 hpi. However, the same group of genes was significantly downregulated in PN at 12 hpi to 48 hpi, indicating that these genes probably participated in plant resistance. Meanwhile, some genes show lower expression levels in PN than in LB at 48 hpi, including VvGATA22, VvGATA6, VvGATA10, VvGATA25, VvGATA24, and VvGATA27, indicating that these genes were likely to take part in plant susceptibility.

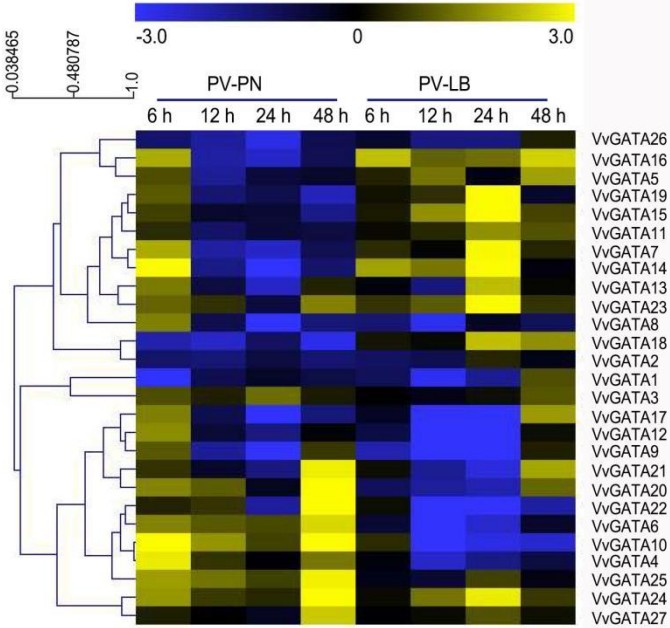

**Figure 5.** Expression patterns of VvGATA factors in response to *Plasmopara viticola*.

Leaves were used to test the expression changes of VvGATA factors at 0, 6, 12, 24, and 48 h after inoculation by *P. viticola* in *V. vinifera* 'Pinot noir' and *V. piasezkii* 'Liuba-8'. The color scale represents expression relative to time 0, with dark blue representing strong down-regulation and yellow representing strong up-regulation. Genes with no significant difference were labeled in black. VvACTIN (AY680701) was used as a reference gene. The relative expression levels were calculated using the $2^{-\Delta\Delta Ct}$ method, and the heat map was visualized using MEV.

### 3.6. VvGATA27 Localizes to the Nucleus and Shows Transcriptional Activity

VvGATA27 transcription factor, which contained the ASXH motif, was first reported in grapes. To assess the transcriptional activation potential of VvGATA27, the CDS from *V. vinifera* cDNA was cloned into the pGBK-T7 vector used in the yeast GAL4 system, and the pGBK-T7-VvGATA27 construction was introduced into the yeast strain Y2H and grown on selective media (SD/-Trp). The clones carrying pGBKT7-VvGATA27 grew well and turned blue on selective single dropout (SD/-Trp) media in the presence of Aureobasidin A (Aba) and the chromogenic substrate (X-α-Gal), indicating that VvGATA27 has transcriptional activity in yeast (Figure 6A). To assess the subcellular localization of VvGATA27, the CDS was cloned into the pCAMBIA2300 vector as a fusion to green fluorescent protein (GFP), and the plasmid was introduced into *V. vinifera* protoplasts using polyethylene glycol (PEG)-mediated transformation. VvGATA27-GFP fusion proteins and GFP (as a negative control), together with NLS-mCherry, which was used as a nuclear-

localized marker, were co-expressed in *V. vinifera* protoplasts. Confocal microscopy revealed that the VvGATA27-GFP fusion protein was localized to the nucleus (Figure 6B).

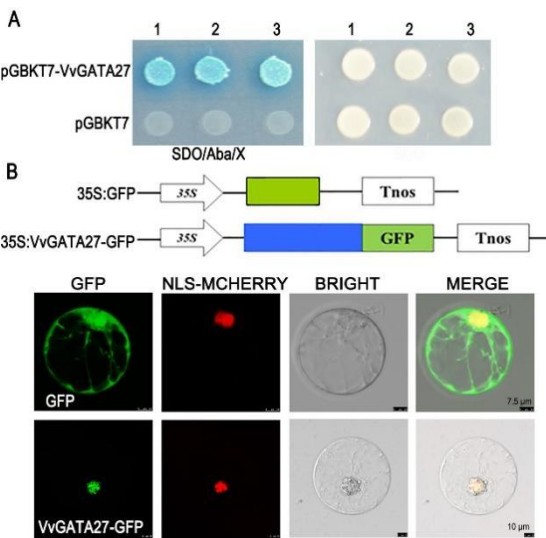

**Figure 6.** Subcellular localization and transcriptional activity of VvGATA27. (**A**) The pGBKT7-VvGATA27 and pGBKT7 (as negative control) constructions were transformed into Y2H gold cells. The yeast assay showed that VvGATA27 has transcriptional activity as evidenced by growth on SDO (minimal media single dropouts, SD-Trp) in the presence of Aba and X-α-Gal, and blue color. The 1, 2, and 3 represent three yeast colonies for each confirmation (**B**) Illustration of the constructions used for the confocal microscopy assay and subcellular localization of VvGATA27 in *V. vinifera* protoplasts. 35S-GFP was used as a positive control. A GFP-VvGATA27 fusion protein and the nuclear RFP fluorescence marker NLS-mCherry were co-expressed in *V. vinifera* protoplasts following polyethylene glycol (PEG)-mediated transformation of the protoplasts. The photos were taken after protoplasts were incubated for 20–24 h in a weak light at 25 °C. Scar bar = 5–10 μm. The experiments were repeated three times with similar results, and at least two protoplasts were observed each time.

## 4. Discussion

In plants, GATA transcription factors have been reported to be involved in light response, development of flowers and chloroplasts, nitrogen metabolism, and response to biotic and abiotic stresses [23,25,27,36]. Our study identified 27 GATA factors in the grapevine genome, adding 8 new genes to the 19 previously identified [27]. Our phylogenetic analysis clearly illustrated that the 27 GATA transcription factors could be classified into 4 subfamilies; this classification is consistent with previous studies in grapevine and *Arabidopsis* [14,24]. To better understand the evolutionary relationships and potential functions of the grapevine GATA transcription factors, the conserved motifs, exon-intron structure, and potential cis-elements were characterized. Members of Subfamily III were highly conserved between grapevine and *Arabidopsis* and possessed a CX2CX20CX2C motif coupled with CCT and Tify (ZIM) motifs, which are characteristic of plant-specific GATA transcription factors [27]. VvGATA27, previously unidentified genes, contained specific conservative motif ASXH belonging to subgroup IV as reported in *Arabidopsis* and pepper [14,34]. In grapevine, GATA transcription factors contain only a zinc finger motif (CX2CX17-20CXC) as described previously [14]. The conservation of protein motifs and exon-intron structures within GATA subfamilies among various plants suggests that members of the subfamilies, at least to a certain degree, have conserved functions. Additionally, the identification of potential cis-elements in the promoters of GATA transcription factor genes suggested potential functions in plant biological processes. Except for the light cis-elements, the cis-elements of MYC and MYB account for a great proportion of the promoter of VvGATA genes as in previous studies in peppers [34]. We identified several SA/JA-response cis-elements such as JERE, as-1, SARE, CGTCA box, and TCA-element,

TGACG-motif, and stress-responsive elements such as LTR, MBS, T/C-rich repeats, and Myb binding site, and even disease resistance-related elements such as the W-box, indicating VvGATAs probable involved in plant defense. Meanwhile, 17 VvGATAs contained cis-elements related to response to SA, and 27 VvGATAs have JA-responsive cis-elements, indicating most genes probable play roles in SA and JA signaling. We found that some VvGATA genes also contain a GATA element, suggesting the possibility that these genes might be autoregulated or regulated by other GATA transcription factors. At present, most GATA transcription factors are reported to be involved in light response and development, but there have been few reports on hormone signaling and pathogen response. To explore the potential roles of grapevine GATA transcription factors in hormone signaling, we examined the expression of VvGATAs in response to SA, MeJA, and BR. We found that a few VvGATAs (VvGATA20, VvGATA26, and VvGATA5 by SA; VvGATA3, VvGATA4, and VvGATA7 by JA) greatly regulated in expression level after 0.5 h treatment, indicating those genes might play important roles in SA and JA signaling. Moreover, the significance of expressions induced by BR was less than by SA and MeJA, indicating that the GATA family gene is probably more sensitive to SA and JA treatment. However, the number of the VvGATA genes that were slightly upregulated at 0.5 h by BR treatment was more than SA and JA. Previous studies have suggested that Arabidopsis GATA2 and GATA4 participate in the BR signaling pathway. In Arabidopsis, BR represses the expression of AtGATA2 via the BR-dependent transcription factor BZR1 [23,37]. Here, we found that the AtGATA2 (At2G45050) ortholog VvGATA24 is significantly downregulated by BR, suggesting that GATA2 and VvGATA24 probably have similar functions. Our results showed that VvGATA genes might also play an important role in hormone response.

The wild Chinese grapevine, *Vitis piasezkii* Liuba-8, shows the strongest resistance to *P. viticola* among any grapevine yet studied [1,4]. To investigate the potential roles of VvGATA genes in downy mildew resistance, we evaluated their expression in response to challenge with *P. viticola*. We found striking differences between Liuba-8 and the susceptible *V. vinifera* cultivar Pinot noir. For example, in Liuba-8, only a few GATA genes were upregulated at 6 hpi, whereas in PN, most of the GATA genes were upregulated by this point, indicating most VvGATA genes were quickly responsive to *P. viticola* in susceptible grapevine genotypes. Previous reports have indicated that those transcription factors that are expressed to higher levels in resistant species are likely to be involved in resistance [25,38]. Compare the differences in the expression patterns in two cultivars; a group of VvGATA genes was upregulated by 12 to 24 hpi and dramatically induced at 24 hpi. However, the same group of genes was greatly downregulated in PN, demonstrating that these genes might act as positive regulators in the immune response to *P. viticola*. Meanwhile, some genes show higher expression levels in PN than in LB at 48 hpi, indicating that these genes were likely to take part in plant susceptibility. These results suggest that GATA genes may participate differently in immunity mechanisms between resistant and susceptible genotypes. In this study, we identified a new GATA transcription factor from grapevines, VvGATA27. VvGATA27 is most closely related in sequence and gene structure to *Arabidopsis* AtGATA26 (AT4G17570) and possessed the special conservative motif ASXH. This motif has not been characterized in grape GATA transcription factors before. Transcription factors such as VvHDZ38 in grapevine and TaGATA1 in wheat are generally located in the nucleus and show transcriptional activity [25,34]. Here, we demonstrated that VvGATA27 is targeted to the nucleus in *Vitis*, and also possesses GATA zinc fingers and shows transcriptional activity. These results suggest its function in grapes as a transcription factor.

## 5. Conclusions

We comprehensively identified 27 VvGATA transcriptional factors in grapevine and characterized phylogenic relationships and cis-elements, speculating that they play roles in plant biological processes. Subsequently, the expression patterns of VvGATA factors in response to hormones and *P. viticola* were investigated, which suggested VvGATAs, might be involved in hormone signal transduction and defense response to downy mildew.

Furthermore, a subfamily IV transcription factor, VvGATA27, targeted the nucleus in *Vitis* and showed transcriptional activity. Our results not only added eight genes to the grape VvGATA family but also laid a foundation for further research on the function of VvGATA transcription factors.

**Supplementary Materials:** The following are available online at https://www.mdpi.com/article/10.3390/horticulturae8040303/s1, Additional File S1, Figure S1: Sequence analysis of the VvGATA27 and AT4G17570, Figure S2: Identification of conserved structural domains within VvGATA proteins, Figure S3: Exon/intron organization of VvGATA family (30 December 2019), Table S1: Table S2 the primers of qRT-PCR, Table More informations of identified GATA genes, Table S3: Proteins sequences of GATA factors, Table S4: cis-elements on the promoter of 27 VvGATA factors.

**Author Contributions:** T.C. and Y.X. designed the research; J.P. performed all gene RT-PCR assays. T.C. and M.L. performed the analyses of subcellular localization and protein interactions. M.L. prepared the plant materials. T.C. drafted the manuscript, M.D., Y.L. and Y.W. modified the language. All authors have read and agreed to the published version of the manuscript.

**Funding:** This work was supported by the National Natural Science Foundation of China (Grants No. 31872054, 31672115 and 31471844) and the Innovation Fund for Young Talents in Fujian Academy of Agricultural Sciences (YC2016-2).

**Institutional Review Board Statement:** Not applicable.

**Informed Consent Statement:** Not applicable.

**Data Availability Statement:** All the data are available.

**Acknowledgments:** The authors thank Steven van Nocker for critically reading the manuscript and his helpful suggestions.

**Conflicts of Interest:** The authors declare that they have no known competing financial interests or personal relationships that could have appeared to influence the work reported in this paper.

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
