# Peer review of "Genetic Analysis of the Grapevine GATA Gene Family and Their Expression Profiles in Response to Hormone and Downy Mildew Infection"

_horticulturae, doi:10.3390/horticulturae8040303_

Round 1

Reviewer 1 Report

 The manuscript titled “Genetic analysis of the grapevine (Vitis vinifera L.) GATA gene family and their expression profiling in response to hormone and downy mildew infection” is devoted to the plant response to  downy mildew, caused by the oomycete Plasmopara viticola. It is known that  members of the GATA family of transcription factors play key roles in light and phytohormone signaling.  Assessment of potential roles for grapevine GATA transcription factors in resistance to downy mildew was done via identification of  GATA transcription transcriptional responses  to three biotic stress-related phytohormones (SA, MeJA, and BR) in callus cells. The transcriptional response to challenge with P. viticola in a downy mildew-sensitive cultivar, V. vinifera ‘Pinot noir’, and a resistant cultivar, V. piasezkii ‘Liuba-8’ was done. Many of the VvGATA genes were expressed at different times or to different levels in grape following hormone treatment or pathogen challenge, suggesting their various roles in plant innate immunity.

Despite the overall good content and writing, many sentences have no clear cense or could be misunderstood. The text must be checked through, and mistyping and errors in style (like missed Italic Latin names) must be corrected.

Abstract, discussion, and conclusions must be written more clearly without duplication of meaning in several sentences.

At the Line 415, Authors wrote: “These results suggest that GATA genes may participate differently in immunity mechanisms between resistant and susceptible genotypes. However, the pattern of GATA genes expression perhaps was  different in many other genotypic features that could be related to P. viticola resistance. But more experiments were needed to verify the conclusion.” 

It is expected that research manuscript describes a theory (idea) or two conflicting theories those are confirmed/denied experimentally. Please, describe clearly your initial idea, and make a clear conclusion on the correctness of it after all experiments were conducted.  

In Abstract, Line 28.

Taken together, the bioinformatics identification and expression profiles of the VvGATA transcription factors in response to hormones and P.viticola, which contribute to increase current knowledge of VvGATA transcription factors in host resistance to P. viticola.

  • Please, re-write and show clearly the main result of the research

Line 37

 GATA transcription factors comprise an evolutionarily conserved family of zinc-finger transcription factors that regulate genes through binding to a defined DNA cis-element [(T/A) GATA (A/G)] in gene regulatory regions [5,6].  

  • Please, re-write to avoid repetition of transcription factors

Line 48

In fungal cells, GATA genes play multiple roles including in nitrogen metabolism, circadian rhythms, and siderophore biosynthesis [11,12].

  • The manuscript is devoted to plants, it is better to concentrate on plant examples, GATA function in fungi and animals can be very different.

Line 69

In fruit fly, GATA transcription factors participate in humoral and tissue-specific immune responses during embryogenesis [23].

  • The same as for Line 48.

Line 146

GLRaV-3 was localised in micrografts after 3 weeks of micrografting. The virus immunolocalisation was described by [12]. The sections were treated with PBS containing  4% BSA for 30 min, followed by overnight incubation at 4 oC with rabbit polyclonal antibodies coat protein to GLRaV-3. After washing three times, the sections were covered with mouse anti-rabbit monoclonal antibodies at room temperature for 30 min. After rinsing three times with PBS again, samples were stained with Fuchsin substrate solution. The sections were observed via a light microscope .

- some words are in blue color, what does it mean?

Line 319

Fig 5 Expression patterns of VvGATA genes in response to Plasmopara viticola.

  • Legend above the figure (scale) show expression change from -3.0 to -3.0. Is it from -3.0 to +3.0.?

Line 354

In plants, GATA transcription factors have been reported to be involved in light  response, development of flowers and chloroplasts, nitrogen metabolism, and response to biotic and abiotic stresses [,22, 25,27,37].

  • Reference numbers start from comma, please, correct.

Line 401 and line 411

  1. viticola must be in italic.

Line 407

Previous reports have indicated that, 407 in response to pathogen in grapevine, those transcription factors that are expressed to 408 higher levels in resistant species are likely to be involved in resistance [25,41].

Line 413

Meanwhile, some genes show lower(Higher)expression level in PN than in LB at 48 hpi, indicating that these gene were likely to take part in plant susceptibility.

  • Please, re-write more clear.

Line 415

These results suggest that GATA genes may participate differently in immunity mechanisms between resistant and susceptible genotypes. However, the pattern of GATA genes expression perhaps was  different in many other genotypic features that could be related to P. viticola resistance. But more experiments were needed to verify the conclusion.

  • What is the meaning of this sentences? You wrote that the results are not clear?

Line 439

we comprehensively identified VvGATA genes in grapevine and characterized  their gene and protein structure and phylogenic relationships.

Line 430

Subsequently, the expression patterns of VvGATA genes in response to hormones and biotic stress treatments were investigated, which suggested important contributions to plant innate immunity through involvement of hormone signal transduction.

  • How VvGATA genes response to hormones suggested their contributions to plant innate immunity?

Author Response

We thank the reviewer for this insightful comment. These comments are absolutely valuable and very helpful for revising and improving our manuscript. We have carefully discussed these comments and have made corrections one by one based on your suggestions. We hope that these changes will be meeting with approval of this journal.

Here we did not list the changes but marked with track changes in revised manuscript.

Reviewer 2 Report

Dear Authors,
Your manuscript presents results of an original, valuable research work. The topic is interesting, the methodology is up-to-date, the results and consequences are correct. However, the structure and implementation of the text should be improved. Please consider the following notes and modify the manuscript accordingly.

Introduction:

In this section please write a paragraph about Vitis piasezkii. This species is not so well-known, however recently it has been extensively studied because of its resistance features (e.g. powdery resistance - Pap et al. BMC Plant Biology, 2016; 16:170). Please show how this species is closely related to V. vinifera. I suggest featuring this species in the title of the paper too.

In the paragraph between line 65 and 73 please delete the second sentence (regarding fruit fly). This part talks about plants, this fruit fly related information is not necessary here.

Methods:

In section 2.3. please use past tense (how DID you the experiment) and don't use imperative mood like in a recipe (see e.g. in ine 116, line 120 etc.).

Please don't use citation in the title line of section 2.4. Furthermore, delete the two sentences in lines 134-137, because they concern infection assay, not the callus culture.

The part between lines 143-152 should be deleted, this text originated from another paper, there is not any relevance here

.Results:

Please don't repeat methodology in this section. E.g., the text between lines 188-194 (section 3.1) is just exactly the same as in section 2.1 (lines 88-99).

General comments:

Please use the scientific names more accurately. At the first mention please write the whole name of an organism (e.g. Vitis vinifera in line 77 and Plasmopara viticola in line 81). Furthermore, scientific names should write italic (e.g. Rhizoctonia cerealis in line 73).

I suggest checking English spelling again, there are lot of mistakes in the text (e.g. leavers instead leaves, line 117, date instead data in line 167, etc.).

Author Response

(The authors gave the same response as above.)
